# Disposable Electrochemical Biosensor Based on the Inhibition of Alkaline Phosphatase Encapsulated in Acrylamide Hydrogels

**DOI:** 10.3390/bios12090698

**Published:** 2022-08-29

**Authors:** Yolanda Alacid, Andrés F. Quintero Jaime, María José Martínez-Tomé, C. Reyes Mateo, Francisco Montilla

**Affiliations:** 1Instituto de Investigación, Desarrollo e Innovación en Biotecnología Sanitaria de Elche (IDiBE), Universidad Miguel Hernández, Avenida de la Universidad s/n, 03202 Elche, Spain; 2Departamento de Química Física and Instituto Universitario de Materiales de Alicante (IUMA), Universidad de Alicante, Carretera San Vicente s/n, 03690 Alicante, Spain

**Keywords:** electrochemical sensor, acrylamide matrix, voltammetric detection, ALP, phosphate inhibitor

## Abstract

The present work describes the development of an easy-to-use portable electrochemical biosensor based on alkaline phosphatase (ALP) as a recognition element, which has been immobilized in acrylamide-based hydrogels prepared through a green protocol over disposable screen-printed electrodes. To carry out the electrochemical transduction, an electroinactive substrate (hydroquinone diphosphate) was used in the presence of the enzyme and then it was hydrolyzed to an electroactive species (hydroquinone). The activity of the protein within the matrix was determined voltammetrically. Due to the adhesive properties of the hydrogel, this was easily deposited on the surface of the electrodes, greatly increasing the sensitivity of the biosensor. The device was optimized to allow the determination of phosphate ion, a competitive inhibitor of ALP, in aqueous media. Our study provides a proof-of-concept demonstrating the potential use of the developed biosensor for in situ, real-time measurement of water pollutants that act as ALP inhibitors.

## 1. Introduction

Alkaline phosphatase (ALP) is a superfamily of very stable and inexpensive metalloenzymes abundantly present in nature. ALP belongs to the group of hydrolases and catalyses the cleavage of phosphoric acid esters releasing the corresponding alcohol and inorganic phosphate. Its active site is electrostatically flexible, with large polar surfaces, which satisfy the requirements for hydrolysis of a wide range of substrates [1]. This promiscuity in substrate recognition and consequently low biocatalytic selectivity, together with its abundance in nature and its physiological role of dephosphorylating compounds, have made ALP activity assessments one of the most widely performed enzyme assays today [2].

ALP is applied in chemical analysis for solving a wide variety of analytical problems. It can be used as a clinical biomarker, since changes in its catalytic activity can be related to pathological situations such as hepatitis, different bone diseases, prostate cancer, etc. It is also widely used in the design of immunosensors and biosensors, either for direct monitoring of analytes (substrates) or for indirect monitoring of inorganic and organic compounds (i.e., heavy metals, herbicides, pesticides, and different biologically active metabolites) which act as inhibitors [3,4,5,6,7]. This second approach is particularly suitable for the screening of drugs or for environmental applications when there is a need for rapid assessment of water quality and real-time warning of water quality decline.

The selection of the substrate for these analytical applications depends mainly on the type of transducer used. Examples of ALP substrates and ALP bioassays strategies include colorimetric, fluorometric, chemiluminescent, and electrochemical techniques among others [8]. Many of these sensing schemes often involve high economic and time costs, as they include several steps and the use of reactants, which can sometimes be very expensive and must be used in a laboratory by specialized personnel. However, for many of the above applications, simpler, cheaper, faster, and more portable devices are needed to enable urgent decisions to be made in real time.

The immobilization of ALP in these sensing devices has the advantage of ease of handling and the possibility of miniaturization, allowing in situ measurements. Silica matrices prepared by the sol–gel process are most often used to immobilize enzymes with very satisfactory results, especially in electrochemical and optical biosensor development [9,10,11]. This methodology has been used in the past to immobilize ALP to develop fluorescent biosensors capable of detecting enzyme inhibitors such as pesticides and phosphate [12,13]. Although the sol–gel process provides a versatile technology, with high biocompatibility and protective properties, one of its main limitations is the low flexibility and extreme fragility (easy breakage) of the matrices, which limits their manipulation, possible miniaturization, and their ability to adapt to any situation and application.

The use of organic hydrogels based on polyimide or methacrylate among other “soft materials” is presented as a very interesting alternative for the manufacture of sensory devices [14,15,16]. These polymeric materials are mechanically stronger and more flexible than sol–gel matrices and can easily accommodate a great number of enzymes, providing a highly aqueous and natural mimicking microenvironment that increases the stability of biomolecules and preserves their functionality [17,18]. In addition, depending on their composition, it is possible to modulate properties such as adhesion, permeability, transparency, degree of swelling, hydrophobicity, conductivity, etc., which makes them easily coupled to different transducers.

Disposable screen-printed electrodes (SPEs) are a convenient research platform for biosensing allowing rapid in situ detection [19,20]. They are usually made by covering a plastic or ceramic support with a conductive paste layer and are often modified with nanomaterials and biomolecules to enhance sensitivity and selectivity. Different electrochemical detection techniques such as chronoamperometry or cyclic voltammetry can be used for analysis; the latter allowing determination of the stability range of the biosensor and the redox processes present on the surface of the modified electrode. Despite the multiple applications of ALP, especially for detecting inhibitors, the modification of SPEs with this enzyme, compared to other enzymes and biomolecules, has been described much less frequently.

In the present work, we have developed an easy-to-use portable electrochemical biosensor based on ALP as a recognition element that can serve as a platform to detect water pollutants that act as ALP inhibitors. Using a green protocol, we have prepared an acrylate-based cationic hydrogel incorporating ALP (ALP@AETAC) that was synthesized directly on a disposable screen-printed carbon electrode. The working principle of the biosensor is based on the detection of hydroquinone, an electroactive species, the product of the hydrolysis reaction of hydroquinone diphosphate (that is non-electroactive) catalysed by ALP. The presence of ALP inhibitors, such as the phosphate ion, reduces the enzymatic activity resulting in a lower production of hydroquinone that can be detected with this device. The device was optimized to detect phosphate in water, a competitive inhibitor of ALP. The determination of this analyte is of great interest, both from a clinical and environmental point of view since variations in its concentrations can be associated with different pathologies and water qualities.

## 2. Experimental Part

### 2.1. Materials and Reagents

Enzyme alkaline phosphatase (ALP, E.C. 232-631-4 from bovine intestinal mucosa, lyophilized powder, ≥10 DEA units mg^−1^, PM 140 kDa), hydroquinone (HQ, ≥97%), [2-(acryloyloxy)ethyl]trimethyl-ammonium chloride (AETAC, 80% wt., 600 ppm monomethyl-ether hydroquinone as a stabilizer), *N*-*N*’-methylene-bis(acrylamide) (MBA, ≥99%), lithium phenyl-2,4,6-trimethylbenzoylphosphinate (LiTPO, ≥95%), and sodium phosphate dibasic (puriss. p.a.) were obtained from Sigma-Aldrich (Merck Life Science, Madrid, Spain). Nitric acid (HNO_3_, 67–70%) from VMR, potassium nitrate (KNO_3_, ≥99%), and tris(hydroxymethyl)aminomethane (Trizma, 99.9%) were used to prepare the buffer solution (0.1 M Trizma, the pH was adjusted to 8.4 by adding nitric acid). Hydroquinone diphosphate (HQ2P), used as enzymatic substrate for the alkaline phosphatase and screen-printed carbon electrodes (SPE, Ref: DRP-C110, geometric area of the working electrode 0.126 cm^2^), were obtained from Metrohm DropSens. All the solutions were prepared using ultrapure water (18 MΩ cm, Merck Millipore^®^ Milli-Q^®^ water, Spain).

### 2.2. Synthesis of ALP@AETAC Hydrogels-Based Electrochemical Biosensor

AETAC hydrogels were synthesized following the slightly modified protocol previously described by Martín-Pacheco et al. [21]. In a typical experiment, 5.66 mL of AETAC (26.4 mmol) and 0.1 g of the crosslinking agent MBA (0.06 mmol) were added to 5 mL of ultrapure water. Finally, the photoinitiator LiTPO (0.02 g, 0.06 mmol) was added to the monomer mixture and the sample was homogenized by gentle agitation in dark conditions for 1 min. For the immobilization of the ALP into the hydrogel, a buffer solution containing the enzyme was mixed with the hydrogel solution in a volume ratio of 1:1 before the crosslinking. For this study, a different concentration of ALP was evaluated, being 6 µM as the optimal enzyme concentration [13].

Before the preparation of the biosensor, the SPE electrode was sonicated in a mixture of ethanol: water (1:1 volume ratio) for 30 min and let dry. Then, a 5 µL-drop of the hydrogel precursor solution (in the absence or presence of the protein) was deposited onto the carbon working electrode of the SPE. Then, the modified electrode was irradiated for 1 min under UV light (*λ* = 335 nm) to initiate the photopolymerization. Figure 1 depicts the stepwise biosensor fabrication.

Therefore, the hydrogel layer is generated in situ on the working electrode, coating it. The adhesion of the matrix to the support is evidenced in the images shown in the Appendix A and in that included in Figure 1.

### 2.3. Electrochemical Measurements

Electrochemical measurements were carried out using an eDAQ Potentiostat (EA163 model) coupled to a wave generator (EG&G Parc Model 175) and the data acquisition was performed with an eDAQ e-corder 410 unit (eDAQ Chart and Scope Software, ES500, Denistone East, Australia). For the electroanalytical and ALP activity measurements, the carbon working-electrode was modified with the 5 µL of the mixture ALP and hydrogel. On top of the hydrogel, a drop volume of 100 µL of trizma buffer solution was added to the electrode surface. Before the addition of the substrate, all the ALP@AETAC-modified electrodes were stabilized by voltammetric cycling between −0.6 and 0.6 V in the buffer solution. Then, an aliquot of the substrate, HQ2P, was added to the drop cell to attain a final concentration of 0.5 mM. For phosphate sensing, the ALP@AETAC-modified electrodes were stabilized in a 100 μL drop in absence of the substrate, followed by continuous addition of different concentrations of phosphate ions up to 10 mM.

## 3. Results and Discussion

### 3.1. Electrochemical Characterization of AETAC and ALP@AETAC-Modified Electrode

Figure 1 shows the stabilized cyclic voltammograms of a bare SPE electrode and an *AETAC*-modified electrode in 0.1 M trizma buffer solution (pH = 8.4) in the absence and presence of 1 mM HQ at a scan rate of 100 mV s^−1^. A stabilized voltammogram was obtained after 8 cycles between the upper and lower potential limits.

A cyclic voltammogram of the modified electrode in the blank solution, Figure 1A, shows an important increase in the double-layer charge contribution as well as the presence of a redox couple at *E*^1/2^ = 0.0 V once the hydrogel is immobilized onto the SPE electrode surface. These peaks are indicative of the presence of electroactive species within the hydrogel, probably due to the presence of stabilizer (hydroquinone species) in the precursor reactant. Considering that the enzymatic reaction of the ALP with the substrate (hydroquinone diphosphate) produces its hydrolysis to hydroquinone (HQ), the electrochemical response of the AETAC-modified electrode has been evaluated in presence of the product HQ.

Figure 1B shows the cyclic voltammogram of 1 mM HQ in solution with the AETAC-modified electrode in trizma buffer. In the scan to positive potentials, an oxidation peak appears at about +0.13 V with a faradaic current intensity of +40 μA, while the counter process appears in the reverse scan at −0.13 V. Peak current values of the oxidation and reduction processes are higher for the modified electrode than the current in the bare SPE electrode, which might suggest a higher electroactive surface area. In addition, the peak separation between the anodic and the cathodic features (∆*E* = 264 mV) shows a more reversible behaviour in the modified electrode than in the bare SPE (∆*E* = 442 mV). The latter could be a result of the ionic conductivity provided by the hydrogel, facilitating the current flow and electron transfer process [22].

Figure 2A shows cyclic voltammograms recorded for an ALP@AETAC-modified electrode in trizma buffer containing the substrate, 0.5 mM HQ2P.

In the initial scan, no clear oxidation or reduction peaks appear related to the presence of HQ, indicating that the substrate lacks intrinsic electrochemical response. In the successive potential cycling, well-defined cathodic and anodic peaks develop located at around −0.02 V and 0.04 V, respectively, in agreement with the formation of HQ species in solution, coming from the substrate hydrolysis catalysed by the encapsulated ALP. These redox peaks grow with the continued cycling reaching a current peak of 37.1 μA for the oxidation process after 152 cycles.

The oxidation peak intensity is directly related to the concentration of product generated (HQ) by the enzymatic reaction. Figure 2B shows the intensity of the oxidation peak as a function of the time, in which a linear correlation between 0 and 20 min can be observed. Considering that initial period, we can determine the activity of the enzyme from the slope of the linear region. Assuming a relation of the HQ concentration with the oxidation peak of 0.025 mM µA^−1^ (as determined in Figure 1B), the hydrolysis rate was calculated to be 0.0235 mM min^−1^ in these conditions. This result suggests that the designed device could be applied to monitor ALP activity in an aqueous environment.

Additional experiments were performed to check the effect of the presence of different anions, other than phosphate, in the biosensor response. We added to the trizma buffer solution chloride, nitrate, and sulfate (sodium salts) anions in concentration 0.05 M in an experiment performed in similar conditions to that shown in Figure 2. The cyclic voltammogram was recorded initially and after 20 min of experiment (see Appendix A). From this voltammogram the hydrolysis rate of the substrate reached was 0.027 mM min^−1^. This rate is very similar to the value in trizma, indicating that these species do not interfere in the response of the biosensor.

### 3.2. Biosensing Assays for Phosphate Detection

To validate the potential application of the ALP@AETAC-modified electrodes as a biosensor platform, the electrodes were tested in the presence of phosphate ions in solution. Figure 3A shows the cyclic voltammograms for the ALP@AETAC-modified electrode in a trizma buffer solution containing 5 mM PO_4_^3−^ + 0.5 mM HQ2P.

The voltammograms show the redox processes associated with the formation of HQ by the enzymatic reaction of the substrate. The current intensity for redox processes grows more slowly in comparison with the same experiments performed in absence of the phosphate (See Figure 2A) Therefore, the presence of PO_4_^3−^ in solution, a well-known inhibitor, induces a decrease in the enzymatic activity.

Figure 3B shows the evolution of the intensity of the oxidation peak related to the presence of HQ as a function of the time for experiments performed at different concentrations of phosphate ion. For the case of the 1 mM phosphate concentration, the hydrolysis rate reaches a value of 0.0198 mM min^−1^, which is a value 15% lower than the hydrolysis rate in the absence of this ion. Higher concentrations of phosphate ions lead to a lower hydrolysis rate, reaching 0.0128 mM min^−1^ and 0.00707 mM min^−1^ for 3.3 mM and 5 mM of phosphate, respectively. The decreasing production rate of the hydroquinone evidences the inhibiting effect and allows the use of this platform as a biosensor to detect this inhibitor molecule in solution.

To optimize the operating conditions for phosphate sensing, a kinetic study of the ALP@AETAC-modified electrode has been carried out in the absence and presence of the inhibitor. We defined a parameter called inhibition current (*IC*) as:$$IC\left(t\right)=I_{p}^{0}\left(t\right)-I_{p}^{\left[{\mathrm{PO}}_{4}^{3-}\right]}\left(t\right)$$
with $I_{p}^{0}\left(t\right)$ being the peak current detected at time *t* in the blank solution and $I_{p}^{\left[PO_{4}^{3-}\right]}\left(t\right)$ the peak current detected at time *t* in the experiment performed in the presence of phosphate. Figure 4A shows the evolution of the peak current related to the oxidation of the product in an experiment performed in the blank solution + 0.5 mM HQ2P and a parallel experiment performed after the addition of 10 mM PO_4_^3−^ blank solution difference of the peak current detected.

The values of *IC*(*t*) are marked as bars for different times, *t* = 10 min and *t* = 30 min. As can be observed, the *IC* values increase with the time of reaction, reaching ~1/3 of the current intensity in absence of the inhibitor after 25 min of measurement.

Figure 4B plots the *IC*(*t*) values as a function of the time of measurement for two experiments performed at different concentrations of phosphate inhibitor (1 and 10 mM). We can observe that the inhibition current increases linearly in both cases for the range time from 0 to 20 min, indicating that the measurement is more sensitive the more time is employed for detection. Measurements performed over 25 min do not produce further enhancement in the *IC* values. In this regard, the sensitivity of the biosensor improves with the time of measurement, thus 25 min is considered an acceptable and optimum time for the use of the biosensor.

Figure 5 shows the evolution of the inhibition current measured after 25 min of incubation—*IC*(25)—for different concentrations of phosphate in the drop electrode.

The addition of PO_4_^3−^ produces a continual increase in the *IC*(25) values due to the rising in the inhibition of the HQ production. The limit of detection for the ALP@AETAC-modified electrode was found to be 0.5 mM of PO_4_^3−^. This value was determined as LOD = 3*σ*/*S*, where *S* = 2.827 µA mM^−1^ is the sensitivity of the sensor, which is the slope of the calibration straight in Figure 5, and *σ* = 0.5 µA is the standard deviation of the blank (hydrogel matrix in the absence of substrate). The developed biosensor showed a broad linear response for phosphate ions in the range of 0–5.0 mM, which together with its detection limit makes this sensor adequate to analyze phosphate in biological fluids or environmental samples [23] with high sensitivity for disposable screen-printed electrodes (22.5 µA mM^−1^ cm^−2^, calculated from the geometric area of the working electrode). In this sense, the proposed biosensor shows competitive performance compared to previously reported work, as is shown in Table 1, offering a portable platform for phosphate detection with a short time of analysis.

It is interesting to highlight that, although enzymatic biosensors with higher sensitivity to phosphate than the one described in this work have been published in the literature, as inferred from Table 1, one of the main advantages of the present biosensor is the voltammetric transduction. Voltammetry makes it possible to detect the presence of other electroactive species that could interfere with the measurements and that can be distinguished from the analyte of interest during the voltammetric sweep. This property is an advantage compared to the usual chronoamperometric responses obtained at a fixed potential, where the measured current can come from different species. This also opens the opportunity to obtain reusable biosensors, one of the objectives we intend to achieve in future work since, after a suitable cleaning procedure, the current related to the entrapped products can be considered as a background current, which can be discarded in subsequent experiments.

## 4. Conclusions

We have developed an easy-to-use portable electrochemical biosensor to measure the activity in situ and in real-time of the enzyme ALP to identify the presence of inhibitors. The biosensor is based on the voltammetry measurement of the electroactive compound hydroquinone, which is the hydrolysis product of the electroinactive substrate hydroquinone diphosphate as catalyzed by the ALP.

AETAC hydrogel containing the enzyme has been directly generated on disposable screen-printed carbon electrodes. The detection of HQ in the hydrogel is more sensitive than the bare SPE electrode. The biosensor was able to quantify the presence of the phosphate ion inhibitor in an aqueous sample with an optimum time of measurement of over 25 min and a LOD of 500 µM. These biosensors can be used to detect and quantify potential ALP inhibitors as potential marine pollutants.

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
