# Peer review of "Disposable Electrochemical Biosensor Based on the Inhibition of Alkaline Phosphatase Encapsulated in Acrylamide Hydrogels"

_biosensors, 2022, doi:10.3390/bios12090698_

Round 1

Reviewer 1 Report

This study describes the development of an easy-to-use, portable electrochemical biosensor based on alkaline phosphatase (ALP) as a recognition element immobilized in acrylamide-based hydrogels using a green protocol and screen-printed disposable electrodes. I believe that revision is necessary to maintain the publication's high quality.

1.     Title of manuscript is too big. It should short and meaningful. Remaing things can be mentioned in abstract. Like, “A portable electrochemical biosensor for detection of phosphate ions”

2.     As mentioned, “In a typical experiment, 5.66 mL of AETAC (26.4 mmol) and 0.1 g of the crosslinking agent MBA (0.06 mmol) were added to 5 mL of ultrapure and deionized water.” Here, need to take 5 mL of ultrapure and deionized water separately? 

3.     As mentioned, “Finally, the photoinitiator LiTPO (0.02 g, 0.06 mmol) was added to the monomer mixture and the sample was homogenized by gentle agitation in dark conditions.” How long need to do this process?

4.     As mentioned, “For this study, different concentration of ALP was evaluated, being 6 μM as the optimal enzyme concentration.” Authors should add the result in support of this 6 uM concentration?

5.     As mentioned, “Then, 5 μl of the hydrogel precursor solution (in the absence or presence of the protein) was deposited onto the working electrode of the SPE electrode and irradiated for 1 minute under UV light (? = 335 nm) to initiate the photopolymerization.” How long need to electrode need to dip into hydrogel solution? Many minor things are missing in this manuscript. That are quite important for beginners who want to do similar works. 

6.     If possible, enhance the visibility and clarity of schematic shown in Fig. 1. 

7.     In Fig. 5, what is the reason for high error bar during measurement?

8.  Authors should also do some more testing like, reusability, repeatability, reproducibility of proposed sensor to showcase the realiability. 

9.     They should also show the spectrum in support of detection time. How much is the detection limit and sensitivity of proposed sensor. 

10.  Authors should show the comparision of performance of proposed sensors with existing sensors to showcase the novelty of work.  

Author Response

Response is attached in the pdf file

Reviewer 2 Report

My main concern in the current manuscript version is the way the LOD was presented (Lines 145-152), more data is needed to conclude that it is 0.5 mM of PO4^3-. Are the values/readings of 0.5 mM phosphate ions statistically significantly different from 1, 5 mM - 10 mM values? 

Second, are there any data coming from environmental samples, be it artificially spiked with phosphate or not, to conclude that the developed biosensor is adequate to analyze phosphate in biological fluids or in environmental samples as stated in Lines 150 - 151?

Author Response

Response in the attached pdf file

Reviewer 3 Report

Alacid et al. demonstrated the electrochemical biosensor for detection of phosphate ions. Authors have explained nicely, however some issues need to be addressed

1.     Author should provide the comparison table of different studies indicating the detection limit and linear range.

2.     Please provide the R2 value of phosphate ion detection.

3.     Author should provide the formula for calculating detection limit.

4.     In order to prove the specificity of sensor, author should perform the experiment with different other ions also.

5.     Author should emphasize more on detection mechanism of biosensor developed.

6.     Please add a section describing the advantage of developed electrochemical biosensor in comparison to other method biosensor.

7.     If possible, author should provide some evidence showing the coating of reagents .

Author Response

Response in the attached pdf file

Round 2

Reviewer 1 Report

Authors have revised the manuscript properly as per suggestions. 

Author Response

Thank you very much for your kind revision.

Reviewer 2 Report

We appreciate the careful reading of the manuscript whose comments and suggestions have improved the quality of the manuscript. Answers to these comments are summarized below. The changes in the revised manuscript were highlighted in yellow. 

My main concern in the current manuscript version is the way the LOD was presented (Lines 145-152), more data is needed to conclude that it is 0.5 mM of PO4^3-. Are the values/readings of 0.5 mM phosphate ions statistically significantly different from 1, 5 mM - 10 mM values? 

The detection limit of the biosensor was 0.5 mM and was calculated, as recommended by IUPAC and described in Thomsen, et al, Spectroscopy, 18: 112-114. 2003, from the following equation: LOD = 3/S, where S=2.827μA mM-1 is the sensitivity of the sensor (the slope of the calibration line inFigure 5) and =0.5 μA is the standard deviation of the blank (hydrogelmatrix in the absence of substrate). The measurements are made in triplicate,each time using a different SPE. This information has been added in themodified version of the manuscript (p 10, line 282). We would like to remarkthat there exist some variabilities related to the real electroactive area ofcommercial SPE. If we normalized the voltammetric measurements to thereal area, less variability between measures would be obtained. It should beremarked we have not made efforts to normalize the response of the differentelectrodes with the real area of each one, since we chose a demandingmethod for the development of the biosensor, considering that all theelectrodes present an equal electroactive area. If the voltammetricmeasurements were normalized to the real area (obtained from the doublelayer charge) the standard deviation obtained would be reduced to one-fifthof the values shown in the article. We emphasize that our initial objectivewas to simplify the measurement to obtain values performing anyrenormalization related to the real area. Although it is usual inelectrochemical measurements, it complicates the use of these sensors inpossible real applications.

Second, are there any data coming from environmental samples, be it artificially spiked with phosphate or not, to conclude that the developed biosensor is adequate to analyze phosphate in biological fluids or in environmental samples as stated in Lines 150 - 151? 

This article shows a proof of concept of a biosensor, and it has not yet been optimized for direct application in real samples. Therefore, all the solutions have been synthetic, and the study was carried out by adding a well-known inhibitor of ALP to the test solution. Although we are conducting a study with possible interferents that may be present in biological samples or in the environment (inorganic ions or various contaminants) and we have observed that they do not affect the response of the biosensor, we want to emphasize that the main objective of this work is not to analyze real samples but to demonstrate that ALP is active after being immobilized in the hydrogel and deposited on an SPE and that it is sensitive to the presence of inhibitors in solution. 

Reviewer's Comments: Thank you for a very forward response to the reviews provided. The reviewer would like to suggest to the authors to add more data from real samples to make the study more meaningful and impactful to the broader audience. Also since the main objective of this work as per the authors' response is only to demonstrate the active state of ALP upon immobilization and deposition, the reviewer is suggesting to update the title as it can mislead readers especially those who are interested to learn novel and new biosensing techniques and tools that are capable of actually detecting target analytes from real samples. The current version is not ready for publication yet. 

Author Response

Reply in the attached file.

Reviewer 3 Report

1. Authors should provide specificity data in manuscript.

2. Author should provide experimental data of coating not schematic diagram.

Author Response

Reply in the attached file.

Round 3

Reviewer 3 Report

1. In the first and second revision, comment is there to provide the specificity data in manuscript but author have not provided it. Authors have penned the following statement “Among others, we have tested chloride, nitrate or sulfate and have observed that these ions, not being recognized as ALP inhibitors, do not modify the response of the biosensor. Some additional experiments have also been performed with emerging contaminants such as iopamidol or glyphosate, but the biosensor was not affected by them” So, in my view they have to provide the data regarding the effect of some ions on the performance of developed sensor. 2. In the first and second revision, comment is there to provide the experimental result to confirm the coating. But authors have provided the schematic diagram which does not anyway confirm the coating. So, due to these two concerns I have rejected the manuscript.

Author Response

According to the reviewer suggestions we modified the manuscript.  As indicated by the reviewer we have included new data on the potential interference by anions in the revised manuscript.

We prepared an additional file with supplementary information including the voltammograms of experiments performed in the presence of chloride, sulfate, and nitrate anions. The voltammograms in the presence of those anions are very similar to those presented in figure 2 and, in order to avoid redundant figures in the main text, we move it to the supplementary data, but we included the information of those experiments in the revised manuscript. In addition, we have included in the supplementary information a series of pictures of the acrylamide coating over the working electrode.